# Human Body Malodor and Deodorants: The Present and the Future

**DOI:** 10.3390/ijms262110415

**Published:** 2025-10-27

**Authors:** Hyun Tae Son, Hyo-Seung Choi, Seung-Sik Cho, Dae-Hun Park

**Affiliations:** 1Department of Veterinary Medicine, Kangwon National University, Chuncheon 24341, Gangwon, Republic of Korea; sht861019@kangwon.ac.kr; 2Department of Digital Contents, Dongshin University, Naju 58245, Jeonnam, Republic of Korea; design@dsu.ac.kr; 3Biomedicine, Health and Life Convergence Sciences, BK21 Four, College of Pharmacy, Mokpo National University, Muan 58554, Jeonnam, Republic of Korea; 4Particle Pollution Research and Management Center, Mokpo National University, Muan 58554, Jeonnam, Republic of Korea; 5College of Korean Medicine, Dongshin University, Naju 58245, Jeonnam, Republic of Korea

**Keywords:** malodor, deodorant, ABCC11 efflux pump, bacterial influx pump, axillary malodor-releasing enzyme

## Abstract

Human axillary malodor negatively influences impression-related appearance, confidence, and hygiene, and ultimately decreases quality of life. Malodor formation involves three steps: vesiculation of odorless precursors within the human body, influx of these precursors into the intracellular space of bacteria, such as *Corynebacterium striatum* and *Staphylococcus hominis*, and efflux of malodorous metabolites into the axilla after conversion by axillary malodor-releasing enzymes (AMREs). Malodor deodorants are currently in use, and their formulation strategies, based on the ingredients, can be classified as follows: anti-sweating, antiproliferation of malodor-forming bacteria, masking (neutralizing) effects against malodor, and deodorization. However, current deodorants have several adverse effects. To reduce such effects while enhancing malodor suppression, a strategy targeting the specific step in malodor formation should be developed, such as the use of ABCC11 pump inhibitors, specific bacterial active pump controllers, and AMRE blockers.

## 1. Malodors in the Body

### 1.1. Definition

Malodor in the human body is a pungent and foul odor from various parts of the body, such as the axilla (axillary odor), mouth (halitosis), and feet (plantar odor) [1]. Malodor is caused by sweating, a physiological process that regulates the body temperature [2]. Sweating begins with the activation of the sweat glands, which can be classified into two types: eccrine and apocrine [3]. Eccrine sweat glands are spread throughout the body, excluding the penis and lips, and have three functions: releasing watery secretions that contain various minerals and organic acids; regulating body temperature; and contributing to the skin barrier’s function [4]. Meanwhile, apocrine sweat glands are located in several areas, such as the axilla, foot, and genitalia, and produce oily fluids containing steroids, fatty acids, and proteins [5]. During the initial release step, sweat is odorless, and malodor formation in the axillary region is induced by normal flora, such as *Staphylococcus* sp. and *Corynebacterium* sp. [6]. Halitosis is caused by malodorous metabolites, such as indole, hydrogen sulfide, and methanethiol, which are produced by anaerobic bacteria in the mouth [7], and plantar odor is caused by metabolites such as methanethiol and isovaleric acid, which are produced via bacterial metabolism [8].

### 1.2. Causes of Malodor Formation

In the 1970s, two steroids, namely 5α-androst-16-en-3-one (androstenone 1) [9] and 5α-androst-16-en-3α-ol (androsenol) [10], were suggested as the main factors in axillary malodor-related studies because of their urine-like smell. However, as 50% or less of population conceived androstenone 1, it was postulated as a primary cause of axillary malodor in 50% or less of individuals [11]. Three carboxylic acids, namely (E)-3-methyl-2-hexenoic acid (3M2H), 4-ethyl-octanoic acid, and 3-hydroxy-3-methyl-hexanoic acid (HMHA), are considered to be important causes of axillary malodors, and their chemical structures are fermented by the zinc-dependent aminoacylase (*Nα*-acyl-glutamine aminoacylase, *Nα*AGA) of *Corynebacterium striatum* Ax20, also called axillary malodor-releasing enzyme (AMRE) [12,13]. Four sulfanylalkanols, 3-sulfanylhexan-1-ol, 2-methyl-3-sulfanylbutan-1-ol, 3-sulfanyl-pentan-1-ol, and 3-methyl-3-sulfanylhexan-1-ol (3M3SH), are fermented by cystathionine-βlyase (C-S lyase) in *C. striatum* Ax20 and are major contributors to axillary malodor [14]. The production of 3M3SH is strongly associated with *Staphylococcus hominis* [15].

#### 1.2.1. Gene Expression

The adenosine triphosphate (ATP)-binding cassette (ABC) transporter superfamily comprises transmembrane proteins that regulates the transport of diverse molecules across plasma membrane using ATP, including ABCA, ABCB, ABCC, ABCD, ABCE, and ABCG [16]. ABCC is classified into 12 members (ABCC1–ABCC12) [17] and is closely related to axillary skin metabolism and malodor [18]. Earwax is composed of ceruminous products secreted by apocrine glands and is classified into wet and dry types. The wet type is strongly related to the malodor of the human body. The single-nucleotide polymorphism (SNP) 538G ⟶ A (rs17822931) in *ABCC11* determines earwax type. The AA genotype, which is prevalent among Northeast Asian populations such as Koreans and Chinese (80–95%), is strongly associated with the dry type. In contrast, the GA or GG genotype, commonly present in Europeans and Africans (97–100%), is associated with the wet type [19]. Harker et al. reported differences in malodor precursors based on the SNP of the *ABCC11* gene, such as AA, GA, or GG, in axillary samples, and found that the levels of malodor precursors such as GlN-HMHA, Gln-3M2H, and Cys-Gly-(S)-3M3SH in the AA genotype were much lower than those in the GA or GG genotype [18].

#### 1.2.2. Diet Consumption

Although some investigations have been carried out, the suggested relationship between food intake and body odor remains unclear. The results regarding the relationship between meat consumption and body malodor are controversial. Havlicek and Lenochova reported that red meat intake was associated with less attractive and pleasant body odor, but the results for non-meat consumption were contradictory [20]. However, Zungia et al. reported that fat and meat intake resulted in a more pleasant body odor while carbohydrate consumption resulted in a less pleasant odor [21]. Garlic consumption is strongly related to body odor attractiveness across both genders; however, the attraction level is based on the dosage of consumption [22]. A double dosage (12 g) of garlic intake, based on the recommended daily amount (6 g), was evaluated through hedonic perception and was suggested to enhance body odor due to its health-promoting properties (antioxidative and antimicrobial effects) [22]. Although no difference in body odor was observed between habitual daily intake and intake restriction for a certain period (48 h) after 72 h, the resumption of food consumption made axillary odor more attractive [23].

#### 1.2.3. Age

Expressions such as “old lady smell,” “old man smell,” or “old person smell are used as idioms to characteristically describe the relationship between odor and age; such idioms suggest that unpleasant body odor increases with increasing age [24] and that the change in odor in the elderly might be related to the decrease in andorogen production [25]. Some volatile compounds, such as dimethylsulfone, benzothiazole, nonanal, and nonenal, are strongly associated with odor in the elderly [26]. Aldehyde 2-nonenals, such as cis-2-nonenal and trans-2-nonenal, which are produced via lipid peroxidation, have unpleasant greasy and grassy odors [27], and volatile compounds, such as dimethylsulfone, benzothiazole, nonanal, and aldehyde 2-nonenal, increase with age, particularly in those aged over 40 years [26,28].

#### 1.2.4. Gender

Studies on the discrimination of odor differences based on gender have been conducted, and several studies that included donors and testers of different genders have reported differentiation rates ranging from 32% to 75% [29,30]. However, few studies have suggested that humans cannot distinguish the differences in odor based on the other gender just judging by odor [31,32]. Most studies on the difference in axillary odor between females and males have reported that the reasons for this are genetic background and hormonal causes, including an axillary odor change based on the phase of menstruation [33].

### 1.3. Mechanism of Malodor Formation in the Axilla

Malodor formation in the body requires two key interactions between humans and the microbiome: the release of odoriferous precursors released from humans through sweat and conversion of the enzymes into malodor by the microbiome (Figure 1).

#### 1.3.1. Sweating and Release by Secretory Vesicles

Sweat glands consist of eccrine and apocrine glands [3] and are regulated by the sympathetic nervous system, particularly acetylcholine [34]. Sweating is controlled by adrenoceptors, cholinergic receptors, and purinoceptors [35]. Some adrenoceptors, such as β-2 and β-3, and some purinoceptors, such as P2Y_1_, P2Y_2_, and P2Y_4_, are present on the apocrine glands, while α-1 adrenoceptors, muscarinic (M_3_) receptors, and purinoceptors, such as P2Y_1_, P2Y_2_, and P2Y_4_, are located on the eccrine glands; however, β-1 adrenoceptors are not present in sweat glands [36]. Hyperhidrosis commonly results in bromhidrosis, and although hyperhidrosis of both eccrine and apocrine sweat glands induces malodor, bromhidrosis occurs more frequently in apocrine than in eccrine glands [4]. Immoderate apocrine secretion stimulates bacterial proliferation, and subsequent bacterial fermentation of the secreted products introduces bromhidrosis. Bromhidrosis is strongly associated with the wet earwax type, which is determined by the AA genotype of *ABCC11* gene [37,38].

Precursors for axillary malodor include carboxylic acids [3M2H, HMHA, 4-ethyl-octanoic acid, and Cys-Gly-(S)-3M3SH], steroids (5α-androst-16-en-3a-ol and 5α-androst-16-en-3-one) [39], and sulfanylalkanols ((S)-3M3SH, 3-sulfanylhexan-1-ol, 2-methyl-3-sulfanylbutan-1-ol, and 3-sulfanyl-pentan-1-ol) [12,13].

For conversion into malodor, the odorless precursors are actively transported into secretory vesicles via the ABCC11 pump and should then be released from the secretory cells into the intercellular space (axillary surface) [40,41].

#### 1.3.2. Transport of Odorless Precursors via Active Transport in the Microbiome

In the intercellular space, secretory vesicles rupture, and the precursors interact with microbes at the axillary surface. In the skin, four genera related to malodor formation have been reported, such as *Propionibacterium*, *Micrococci*, *Staphylococcus*, and *Corynebacterium,* with *Staphylococcus* and *Corynebacterium* being strongly associated with axillary malodor [1]. HMHA and Cyc-Gly-(S)-3M3SH are representative malodors in the axilla; HMHA is produced by *C. striatum* Ax20 [14], and Cyc-Gly-(S)-3M3SH is converted by *S. hominis* [15,42], respectively. Released precursors that interact with microbiomes are taken up by microorganisms via active transporters. The peptide transporter *S. hominis* (PepT_sh_) is known as an active pump for precursors in *S. hominis*, whereas the active transporter of precursors for *C. striatum* Ax20 is currently unknown [39].

#### 1.3.3. Fermentation of Precursors via Converting Enzyme and Malodor Diffusion

Malodor compounds are metabolites of precursors that microbes convert or ferment after obtaining nutrients. Several enzymes (AMREs) convert precursors into malodors, such as N_α_-AGA, thiol precursor dipeptidase A (TpdA), and cysteine β-lyase of *C. striatum* Ax20 [39] and PatB cysteine-S-conjugate β-lyase of *S. hominis* [43]. Finally, the malodor produced by the microbiome passively diffuses to the axillary surface and can be detected by the olfactory system. Table 1 presents a summarization of the precursors, metabolites, bacteria and enzymes for conversion from precursors to metabolites, and odors.

## 2. Deodorants

### 2.1. Definition

A deodorant is a counteracting substance against body odor, such as the axilla, feet, mouth, and other sites [44]. As the US Food and Drug Administration (FDA) classifies deodorants as over-the-counter drugs and the European Union (EU) categorizes them as cosmetics, their safety and efficacy should be definitively confirmed.

### 2.2. Ingredient Classification Based on the Functions

The ingredients in deodorants and antiperspirants include triethyl cytrate, alcohol, ethylhexylglycerin, caprylyl glycol, and potassium alum, while the most important functional ingredients of antiperspirants are aluminum-based salts, such as aluminum chlorohydrate, aluminum sesquichlorohydrate, and aluminum chloride [50]. According to Kalinowska-Lis et al., almost all deodorants and antiperspirants include fragrance-related ingredients, such as parfum, limonene, linalool, citronellol, citral, benzyl salicylate, hexyl cinnamal, and geraniol [50]. Accordingly, the ingredients can be classified as having different effects: anti-sweating, antiproliferation of malodor-forming bacteria, masking/neutralizing effects against malodor, and inhibition of microbial conversion enzymes (Table 2).

#### 2.2.1. Anti-Sweating Effect

As an initial step in the production of body malodor, odorless precursors within secretory vesicles should be released, and if substances that can block the secretion of precursors are used, the conversion of these precursors into malodor can be fundamentally eliminated. The main purpose of antiperspirants is to inhibit the secretion of sweat from sweat gland pores via gel plugging [78], and several types of anti-sweating materials have been typically used, such as aluminum-based compounds [79].

Aluminum-based compounds (aluminum chlorohydrate, aluminum sesquichlorohydrate, aluminum bromide, aluminum zirconium tetrachlorohydrate, aluminum lactate, and potassium alum) have been used as anti-sweating ingredients in antiperspirants since the introduction of aluminum chloride as an ingredient of antiperspirant [51]. However, their safety has been questioned, as repeated application on the body can lead to accumulation in sites such as the epidermis, dermis, subcutis, and lymph system and cause various problems such as hair loss and breast cancer [57,80,81].

Cyclomethiocone is a mixture of cyclic dimethyl polysiloxane compounds such as cyclotetrasiloxane (D_4_), cyclopentasiloxane (D_5_, decamethylcyclopentasiloxane), cyclohexasiloxane (D_6_), and cyclohepatiloxane (D_7_), which are soluble and safe in various solvents such as ethanol, isopropanol, mineral oil, paraffin, and stearyl alcohol, excluding water, and are widely used in many products such as sun protectors, lubricants, and personal care products including antiperspirants [53,82]. Although it has already been approved for use in various personal care ingredients by the US FDA, its potential accumulation in the environment is becoming an issue owing to its ease of being washed off and drained after personal application [53].

Stearyl alcohol has been used as an emollient, emulsifier, and thickener for cosmetics since safety reports; it is hydrophobic and can form a film on the axilla [54]. Castor oil is a natural oil that is extracted from seeds of *Ricinus communis* (Euphorbiaceae) [55], and hydrogenated castor oil (HCO), also called castor wax, is made by adding hydrogen to castor oil. As the main characteristic of HCO is hydrophobicity, most products that include it are waterproof, such as lipsticks, eyeshadows, and antiperspirants [56]. A case of axillary dermatitis caused by HCO was reported in 2008 [82].

#### 2.2.2. Antiproliferative Effect on Malodor-Forming Bacteria

Triclosan (2,4,4′-trichloro-2′-hydroxydiphenyl ether), also known as Irgasan, Irgacare, or Lexol, is a synthetic agent with broad-spectrum antimicrobial effects [57]. It has antimicrobial effects and has been used as an ingredient in personal care products such as soap, shampoo, cream, lotion, and deodorant [83,84]. In 1996, Bhargava and Leonard reported the safety of triclosan in terms of acute, subacute/subchronic, and chronic toxicities [85]. Meanwhile, possible triclosan metabolite toxicity in the environment has been reported [86].

Essential oil, which is extracted from various plants through steam distillation, is an aromatic volatile liquid that has antimicrobial and anticancer properties. Its major components are terpenes, alcohols, esters, and ketones [57]. Based on its biological properties, it has been used as an ingredient in cosmetics and alternative medicine, especially as it can effectively eliminate ammonia odor via its bacteriostatic or bactericidal effects against ammonia-producing bacteria and is used as a deodorant [1,44]. Although essential oils from natural products are assumed to be safe and non-toxic compared with synthetic materials, thorough consideration should be given, as some of the ingredients in essential oils can cause allergies [58] (Table 3).

In 2017, Traupe et al. evaluated the malodor-reducing effect of polymeric quaternary ammonium compounds (PQ-16, copolymers of 1-vinyl-2-pyrolidone and 1-vinyl-3-methylimidazolium chloride) as a formula in a roll-on deodorant and confirmed their antibacterial activity as well as the effectiveness of PQ-16 as a commercial deodorant [59].

Deodorants, including those with aluminum chlorohydrate, completely inhibit the proliferation of microbes such as *Staphylococcus epidermidis*, *Staphylococcus aureus*, *Candida albicans*, *Streptococcus pneumoniae*, and *Escherichia coli*. Aluminum chlorohydrate is the best component for suppressing malodor formation, which is caused by bacterial reactions, as it can eliminate normal flora in the skin; however, the vulnerability of the body area to which it is applied should be considered [60].

#### 2.2.3. Masking/Neutralizing Effect Against Malodor

Fragrance is an important ingredient in deodorants and antiperspirants because it can eliminate malodors [50]. Two major factors are required for an effective fragrance as a deodorant ingredient: substantivity on applied body areas, such as the foot and axillae, and the efficacy of malodor coverage [87]. Many materials have been used as fragrances for deodorant ingredients, such as linalool (3,7-dimethyl-1,6-octadien-3-ol), citronellol, citral, benzyl salicylate, hexyl cinnamal, geraniol, limonene, farnesol, florhydral, and linalyl acetate. Linalool, which is a fragrance contained in herbs, leaves, flowers, and wood (e.g., ho leaf, bois-de-rose, coriander, linaloe, lavender, petit grain, and bergamot), is broadly used as an ingredient in many products such as deodorants, beverages, hard and soft candies, chewing gum, ice cream, and meat products, owing to its fruity flavors such as citrus [61,88].

Essential oils containing citronellol, a monoterpene alcohol, have been traditionally used as a culinary material and medicine [89], and 3,7-dimethyl-6-octen-1-ol (dl-citronellol) is used in many products such as beverages, chewing gum, frozen dairy, gelatin and pudding, and hard candy [62]. Acyclic monoterpene aldehyde (citral) is extracted from many plants such as Lemon myrtle, *Litsea citrata*, *Litsea cubeba*, *Cymbopogon citratus*, and *Leptospermum petersonii*, which have traditionally been used in food and medicine [90], and continues to be a useful ingredient in many industries such as food, cosmetics, and pharmaceuticals [63]. 2-hydroxybenzoic acid (benzyl salicylate) has been widely used as a fragrance ingredient for cosmetic and non-cosmetic products such as decorative cosmetics, shampoo, toilet soap, household cleaner, and detergents. Lapczynski et al. conducted a toxicological and dermatological review of benzyl salicylate and reported its relative safety for use in commercial products in these industries [64]. 2-hexyl-3-phenyl-2-propenal (hexyl cinnamal) is widely used as a fragrance ingredient in the cosmetic industry and has been categorized as a weak skin sensitizer through a local lymph node assay, guided by the Organization for Economic Cooperation and Development [65]. (2E)-3,7-dimethylocta-2,6-dien-1-ol (geraniol), an essential oil present in many natural products such as *Cananga odorata* Hook, *Citrus bergamia* Risso, *Citrus limon*, *Citrus sinensis*, and *Citrus paradisi*, is widely used as a fragrance with fresh, sweet, and rose-like scents [66] but is possibly allergenic [58]. 1-methyl-4-(1-methyl phenyl) cyclohexene (limonene), a monocyclic terpene, has isomers such as R-(+)- and L-limonene, which are obtained from many plants [91]. R-(+)-Limonene is present in steam-distilled essential oils from the peel of *C. sinensis* [92]. Owing to its lemon-like scent, R-(+)-limonene has been used in many industries such as cosmetics, beverages, and foods; however, because it has several toxicities such as hyaline droplet nephrotoxicity in male rats, hepatotoxicity, and neurotoxicity, it is classified as a low-toxicity material [67]. 3,7,11-trimethyl-2,6,10-dodecatrien-1-ol (farnesol) has been widely used for many applications, such as in cosmetics, fragrances, shampoos, toilet soap, and household cleaners, but has toxicities including skin irritation and weak sensitization [68]. Isopropylphenylbutanal (florhydral) is a synthetic compound that has a natural odor similar to that of lily of the valley and hyacinth and has exhibited minor toxicity upon repeated dose exposure (margin of exposure > 100); therefore, it has been used to produce several goods, such as air fresheners, laundry detergent, and fragrances [69,93]. 3,7-dimethylocta-1,6-dien-3-yl acetate (linalyl acetate), which is extracted from *Lavandula angustifolia* essential oil, has been used in many products, such as decorative cosmetics, shampoos, toilet soaps, household cleaners, and detergents, but can cause skin and eye irritation [70]. Linalyl acetate, also known as bergamol, has antidepressant effects and has thus been used as a material for aromatherapy [94].

#### 2.2.4. Deodorizing Effect of Natural Products

Hydrogen sulfide and methanethiol are major malodorous gases [95]. Green tea extract, eracidated hydrogen sulfide, and polyphenol oxidase and peroxidase remove thiols, with mixed formulas of green tea and materials containing polyphenol oxidase and peroxidase exerting more potent effects on malodors, such as hydrogen sulfide and thiols [96]. *Eriobotrya japonica*, an evergreen tree, is distributed in Northeast Asian countries including Korea, China, and Japan and has been used as a traditional medicine for coughing and indigestion in cosmetics owing to its anti-inflammatory and antiallergenic effects [97]. *E. japonica* seed oil contains several fatty acids, such as linoleic, palmitic, behenic, and lignoceric acids, which eliminate the odor of allyl methol sulfide and are therefore used as deodorizing materials [92]. Lotus corniculatus possesses antimicrobial, antiparasitic, and anticancer properties [98,99,100]. *Lactobacillus acidophilus* KNU-02-related bioconverted seeds eliminate *Corynebacterium*- and *Anaerococcus*-produced malodor [72]. *Salvia officinalis* (sage), which has been used as a culinary material and traditional medicine for the treatment of several diseases such as seizure, ulcer, gout, and rheumatism, contains many compounds such as arabinose, borneol, camphor, caryophylene, cineole, elemene, galactose, glucose, humulene, ledene, mannose, pinene, xylose, and uronic acids [101]. Uronic acids in sage control malodor by suppressing the proliferation of *Corynebacterium* and *S. epidermidis* [73]. *Rosmarinus officinalis* L. (rosemary) has been used in traditional medicine for the treatment of various disorders such as those of the musculoskeletal, gastrointestinal, respiratory, circulatory, nervous, and dermal systems and has also been used as incense [102]. Recently, it has been utilized as a deodorizing agent for fish via phenolic compounds present in rosemary, such as rosmarinic acid, carnosic acid, and carnosol [74]. *Lavandular angustifolia* Mill. (lavender) has been used for cosmetic and medicinal purposes since the Greek and Roman era. Recently, it has been used for many purposes such as personal care products (soap, scrub, shower gel, lotion, cream, and face mask), culinary products (honey, tea, and sugar), household items (candle and detergent), and medicine, owing to its many biological effects, including deodorizing, sleep induction, psychological stabilization, anti-inflammatory, antioxidant, and antimicrobial effects, which are mediated by effective constituents such as linalool, linayly acetate, camphor, α-linonene, and camphene [75]. In addition to its use as a deodorizing agent in patients with ostomies, lavender helps to improve their quality of life [76]. Thyme herbs are used in cosmetic, culinary, and medicinal applications [40]. The two most common thyme herbs, *Thymus vulgaris* L. and *Thymus zygis* L., have recently been used for antibacterial, antifungal, and antiviral purposes, as they have many components including the isomeric phenolic monoterpenes thymol (2-isopropyl-5-methylphenol), carvacrol (2-methyl-5-(propan-2-yl)phenol), p-cymene, γ-terpinene, linalool, β-myrcene, and terpinen-4-ol [77].

## 3. Intervention Strategy for Next-Generation Deodorants

### 3.1. ABCC11 Pump Controller

The ABCC11 (multidrug resistance protein 8, MRP8) gene encodes an efflux pump that plays the most important role in regulating malodor and cerumen synthesis. A significant individual difference in the quantity of volatile organic compounds (VOCs) has been observed based on ethnicity [19], meat consumption (although there are controversial results [20,21]), and age [24,26], but not in axillary sweat and cerumen [103,104]. Several compounds from natural products have been reported to inhibit ABCC11-mediated vesicle transport, including genistein from soybean (Glycine max), water extract, phloretin, luteolin, nobiletin, myricetin, quercetagetin, and isoliquiritigenin, with the latter five resulting in complete inhibition [105]. MK-571 [5-(3-(2-(7-chloroquinolin-2-yl)ethenyl)phenyl)-8-dimethylcarbamyl-4,6-dithiaoctanoic acid sodium salt hydrate] is an MRP-specific inhibitor that includes MRP8 (ABCC11 efflux pump) [106].

Miura et al. (2007) reported that this gene could be excluded from the long-established correlation of the ABCC11 gene with the breast via comparing the difference in the frequency of non-colostrum and the volume of colostrum between a dry earwax group (AA genotype) and wet group (AG and GG genotype, wild type) [107]. As the ABCC11 efflux pump secretes molecules from the cytoplasm to the intercellular space, the wild type of ABCC11 is strongly related to resistance to cancer drugs such as 5-fluorouracil (5-FU), 5-fluoro-2′-deoxyuridine 5′-monophosphate (FdUMP), cytosine arabinoside (Ara-C), pemetrexed (MTA), methotrexate (MTX), and 9′-(2′-phosphonyl-methoxyethyl)adenine (PMEA) [108].

### 3.2. Bacterial Active Pump Controller

Proton-coupled oligopeptide transporters (POTs) are transmembrane active pumps responsible for the cellular uptake of di- and tripeptides and prodrugs in eukaryotes and prokaryotes. Different POTs exhibit multiple mechanisms of proton coupling [109], which may enable the development of specific bacterial POT regulators. *S. hominis* is one of the two major bacteria that cause malodor in the axilla, and it utilizes a specific POT, PepT_Sh_, to import malodor precursors such as 3M3SH, which are released from secretory cells into the inner space of bacteria. An important aspect of the relationship between malodor precursors, 3M3SH, and PepT_Sh_ is that water is required to facilitate their binding [47].

Mammalian POTs are called solute carrier (SLC) transporters or the POT family (SLC15), and SLC15 has four types: SCL15A1 (peptide transporter 1, PepT1), SCL15A2 (peptide transporter 2, PepT2), SCL15A3 (peptide/histidine transporter 2, PhT2), and SCL15A4 (peptide/histidine transporter 1, PhT1). In particular, SLC15 is involved in the absorption and excretion of agents, such as aminocephalosporins, angiotensin-converting enzyme inhibitors, and antiviral prodrugs [110].

### 3.3. Bacterial Converting Enzyme Modifier

To produce malodor in microbes, odoriferous precursors should be converted or fermented by AMREs such as N_α_-AGA, TpdA, and C-S lyase [111], and whether AMRE modifiers can block malodor formation should be considered.

*C. striatum* Ax20 converts odorless precursors into malodorous compounds (3-methyl-2-hexenoic acid) using N_α_-AGA [13]. Gln-carbamate, which was synthesized by Natsch et al., suppressed malodor production by controlling N_α_-AGA activity, acting as a competitive substrate of N_α_AGA, and phosphoric acid-derived inhibitors acted as AMRE modifiers via the same pathway as Gln-carbamate [42,112]. Although carbamate does not inhibit AMRE activation, it acts as an AMRE modifier by replacing the precursors for malodor formation and producing fragrant alcohol. Phosphinic acid-derived inhibitors control AMRE activation by altering the chemical structure of AMRE; as AMRE becomes very stable, it cannot convert the precursor into its malodorous form.

Dipeptide-contained Cys-Gly-(S)-3M3SH and Cys-(S)-3M3SH are thioalcohol odor precursors. TpdA cleaves Cys-Gly-(S)-3M3SH to produce Cys-(S)-3M3SH, and only C-S lyase cleaves Cys-(S)-3M3SH in *C. striatum* Ax20 and *S. hominis* [111,113]. C-S lyase has been used as a catalyzing enzyme of β carbon-sulfur bonds to produce flavor compounds in various food manufacturing processes [14]; however, it is also involved in malodor production in the human body by *S. hominis* and *C. striatum* Ax20, which produce malodorous sulfanylalkanols using cysteine β-lyase (cystathionine β-lyase) [114]. ο-Phenanthroline is a TdpA inhibitor that blocks the cleavage of Cys-Gly-(S)-3M3SH [111], and tannic acid suppresses the ability of C-S lyase to cleave Cys-(S)-3M3SH [115].

The bactericidal effect of 4.5% zinc glycinate solution was low, similar to that of 5% aluminum chlorohydrate; however, zinc glycinate effectively controlled bacterial exoenzymes, such as aryl sulfatase and β-glucuronidase, and was used as a deodorant component [116]. 16,5α-androstene-3β-ol and 16,5α-androsten-3-one are malodor steroids and are generated by hydrolytic enzymes such as aryl sulfatase and β-glucosidase [117]. A trial to investigate inhibitors against these two enzymes had been conducted [118].

*S. epidermidis* and *S. aureus* ferment sodium L-lactate to produce malodorous metabolites, such as pyruvate and diacetyl (2,3-butanedione), by changing the intracellular metabolic flow [119,120]. 5-methyl furfural regulates malodor metabolite generation by blocking the activity of acetolactate synthase (ALS) [118], and *Glycyrrhiza glabra* root extract and α-tocopheryl-L-ascorbate-2-O-phosphate diester potassium salt effectively inhibit diacetyl synthesis [120].

In Figure 2, the intervention strategy for next-generation deodorants is summarized.

## 4. Discussion

Malodor of the human body decreases the quality of life of individuals, as it causes a negative impression regarding appearance, confidence, and hygiene [121]. Accordingly, the global deodorant market is forecasted to rapidly increase from USD 26.96 billion in 2024 to USD 42.19 billion in 2032 (5.81% compound annual growth rate) [1].

Malodor formation in the axilla comprises three steps. The first step involves the formation of odorless precursor-containing vesicles in the epithelial cells of the body, such as those in the axilla, mouth, and foot [1]. The ABCC11 efflux pump is closely related to vesiculation [18]. Based on the genotype, the level of vesculation is determined by the difference in malodor levels according to ethnological variation [19]. The second step is the entry of the released odor precursors, including sweat, from the axilla into the intraspace of bacteria, such as *C. striatum*, *S. hominis*, and *S. aureus*, through POTs such as PepT_Sh_ [12,13]. The third step is the metabolism of odor precursors by bacterial AMREs, such as N_α_AGA, TpdA, C-S lyase, aryl sulfatase, β-glucuronidase, and ALS [13,111,113,116,119]. The compounds metabolized by AMREs include 3-methyl-2-hexenoic acid, steroids (5α-androst-16-en-3-one and 5α-androst-16-en-3α-ol), carboxylic acids (HMHA, 3M2H, and 3M3SH), pyruvate, and diacetyl [9,10,12,13,15,119,120].

The present strategies for malodor suppression can be classified as anti-sweating, antiproliferation of malodor-forming bacteria, masking (neutralizing) effects against malodor, and deodorization, and they are utilized in deodorant manufacturing. Antiwetting ingredients include aluminum-based compounds, cyclomethiocone, stearyl alcohol, and castor oil [51,53,59]. The antiproliferative components of malodor-forming bacteria include triclosan, essential oils, polymeric quaternary ammonium, and aluminum chlorohydrate [57,59,60]. Fragrances are representative masking (neutralizing) agents against malodor, and they include linalool, di-citronellol, citral, benzyl salicylate, hexyl cinnamal, geraniol, limonene, farnesol, floryhydral, and linalyl acetate [61,62,63,64,65,66,67,68,69,70]. Because natural products have many biological effects, such as antibacterial, malodor-masking, and deodorizing effects, they have been used as ingredients in deodorants [71,72,73,74,76,77]. However, many materials used as deodorant ingredients have adverse effects, such as the possible induction of breast cancer [81], accumulation in the environment [53], the induction of axillary dermatitis [63], general toxicity [86], induction of allergy [58], increased skin vulnerability [60], skin irritation/sensitization [58,68], and organ toxicity [67].

To avoid such adverse effects of current deodorant ingredients, next-generation deodorants should be suggested based on their appropriate functions at each step of malodor formation. The first-step inhibitors that control ABCC11 efflux pumps include luteolin, nobiletin, myricetin, quercetagetin, isoliquiritigenin, and MK-571 [105,106]. Meanwhile, second-step regulators that suppress bacterial active pumps are difficult to develop because of the high similarity between bacterial and mammalian POTs such as PepT1, PepT2, PhT1, and PhT2 [109,110]; however, it is necessary to develop a specific controller for the bacterial active pump. The third-step suppressors that block the activity of AMREs include Gln-carbamate against NαAGA [113], ο-phenanthroline against TdpA [Emter, 2008], tannic acid against C-S lyase [115], zinc glycinate against aryl sulfatase and β-glucuronidase [116], and 5-methyl furfural against ALS [119].

The traditional methods with which to produce deodorants have focused on altering the environmental conditions that favor bacteria proliferation and activation [51,52,53,54,55,56,57,58,59,60] or on masking malodor [61,62,63,64,65,66,67,68,69,70]. For example, the purpose of anti-sweating functions is to eliminate humid conditions for bacteria [51,52,53,54,55,56], the purpose of anti-proliferation of bacteria is to prohibit bacterial proliferation using antibacterial materials [57,58,59,60], and the purpose of masking properties/those that neutralize malodor is to hide malodor using with fragrances but not to remove the fundamental causes [61,62,63,64,65,66,67,68,69,70]. Although in the past the materials for making deodorant have been used from chemical synthetics, recently, many trials have been run to find them from natural products, as these natural products are supposed to be safer than chemicals [71,72,73,74,75,76,77]. Nevertheless, traditional methods to limit the conditions for bacterial growth might be powerful for eliminating malodor as beneficial microorganisms exist on the body surface, and the traditional methods can inhibit their action, requiring safer and more effective materials [1,44]. As the purpose of the methods required to mask/neutralize malodor is just to conceal it, repeated application can aggravate the malodor generation environment [44,122].

The intervention strategy for next-generation deodorants still focuses on fundamentally preventing malodor generation, but it is safer for the human body as it does not eliminate the environmental conditions required for bacteria proliferation, does not result in a dramatic increase in malodor-generative bacteria unlike the case for fragrances, and inhibits the gathering of precursors in secretory vesicles by controlling the ABCC11 pump [106,107,108] in order to introduce released precursors to the bacteria [110], and to activate malodor-generative bacterial enzymes [42,111,112,115,116,117].

## Figures and Tables

**Figure 1 ijms-26-10415-f001:**
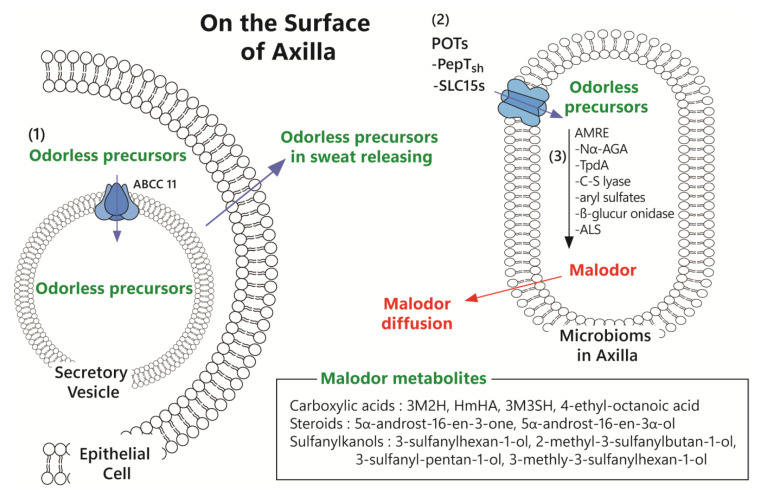
Malodor formation in the axilla. (1) Odorless precursors are transported into secretory vesicles via the ABCC11 efflux pump in secretory cells and are then released onto the axillary surface. (2) After release, the precursors interact with skin microbiota and are taken up by *Staphylococcus hominis* through active transporters such as the peptide transporter *S. hominis* (PepT_sh_); however, the active transporter in *Corynebacterium striatum* Ax20 for such precursors has yet to be identified. (3) The bacteria utilize these precursors as nutrients, converting/fermenting them via several enzymes such as Nα-acyl-glutamine aminoacylase (Nα-AGA), thiol precursor dipeptidase A (TpdA), and β-lyase of *C. striatum* Ax20 and PatB cysteine-S-conjugate β-lyase of *S. hominis*. 3M2H, (E)-3-methyl-2-hexenoic acid; HMHA, 4-ethyl-octanoic acid and 3-hydroxy-3-methyl-hexanoic acid; PepT_sh_, peptide transporter *S. hominis*; AMRE, axillary malodor-releasing enzyme; Nα-AGA, Nα-acyl-glutamine aminoacylase; TpdA, thiol precursor dipeptidase A.

**Figure 2 ijms-26-10415-f002:**
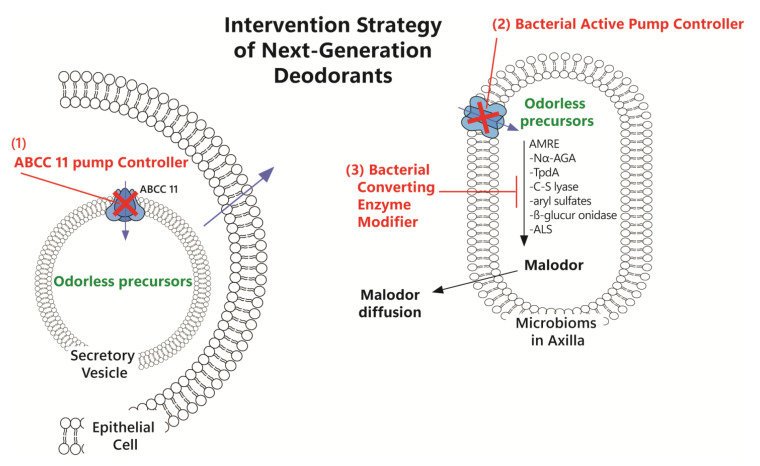
Diagram of the intervention strategy for next-generation deodorants. Purple arrows mean the movement of odorless precursors. 3M2H, (E)-3-methyl-2-hexenoic acid; HMHA, 4-ethyl-octanoic acid and 3-hydroxy-3-methyl-hexanoic acid; PepT_sh_, peptide transporter *S. hominis*; AMRE, axillary malodor-releasing enzyme; Nα-AGA, Nα-acyl-glutamine aminoacylase; TpdA, thiol precursor dipeptidase A.

**Table 1 ijms-26-10415-t001:** Odorless precursors and odoriferous metabolites.

Precursor	Metabolite	Classification	Key Bacteria	Enzyme	Odor	Reference
Nα-3M2H-glutamine conjugate	3M2H	Carboxylic acid	*Corynebacterium striatum* Ax20	Nα-acyl-glutamine amonacylase (N-AGA)	Cheesy/sweaty	[13,14]
Nα-HMHA-glutamine conjugate	HMHA		*Corynebacterium* spp.	Nα-acyl-glutamine amonacylase (N-AGA)	Acidic, pungent	[14,44]
Nα-4-ethyloctanoyl-glutamine conjugate	4-ethyl-octanoic acid		*Corynebacterium* spp.	Nα-acyl-glutamine amonacylase (N-AGA)	Goaty	[14,45]
Glutathione-pathway dipeptide	Cyc-Gly-(S)-3M3SH		*Staphylococcus hominis*	C-S lyase PatB PepT_sh_ peptidases	-	[42,46,47]
Not known as amino-acid conjugate	5α-androst-16-en-3α-ol (androstenol)	steroid	-	-	Musky	[48]
Not known as amino-acid conjugate	5α-androst-16-en-3-one (androstenone)		-	-	Urine/musky	[48]
(S)-3M3SH	Cys-Gly-(S)-3M3SH	sulfanylalknol	*Staphylococcus* spp.	PatB-like C-S lysase	Sulfury, very pungent	[42,44,49]
Cysteine/dipeptide conjugate	3-sulfanylhexan-1-ol (3SH)		*Staphylococcus* spp.	PatB-like C-S lysase	Grapefruit	[14,49]
Cysteine/dipeptide conjugate	2-methyl-3-sulfanylbutan-1-ol		*Staphylococcus* spp.	PatB-like C-S lysase	Onion	[14]
Cycteine/dipeptide conjugate	3-sulfanyl-pentan-1-ol		*Staphylococcus* spp.	PatB-like C-S lysase	Sulfury	[14]

**Table 2 ijms-26-10415-t002:** Deodorant ingredient classification based on function.

Function	Ingredient	References
Anti-sweating	Aluminum-based compounds (aluminum chlorohydrate, aluminium sesquichlorohydrate, aluminium bromide, aluminium zirconium tetrachlorohydroate, aluminium lactate, and potassium alum)	[51]
	Cyclomethiocone (mixture of cyclotetrasiloxane (D_4_), cyclopentasiloxane (D_5_), cyclohexasiloxane (D_6_), and cyclohepatiloxane (D_7_))	[52,53]
	Stearyl alcohol	[54]
	Castor oil/hydrogenated caster oil (HCO)	[55,56]
Anti-proliferation of malodor-forming bacteria	Triclosan (2,4,4′-trichloro-2′-hydroxydiphenyl ether)	[57]
	Essential oil	[57,58]
	polymeric quaternary ammonium compound (PQ-16, copolymers of 1-vinyl-2-pyrolidone and 1-vinyl-3-methylimidazolium chloride)	[59]
	aluminum chlorohydrate	[60]
Masking/neutralizing effects against malodor (fragrance)	Linalool (3,7-dimethyl-1,6-octadien-3-ol)	[61]
	*dl*-citronellol (3,7-dimethyl-6-octen-1-ol)	[62]
	Citral (acyclic monoterpene aldehyde)	[63]
	Benzyl salicylate (2-hydroxybenzoic acid)	[64]
	Hexyl cinnamal (*2*-hexyl-3-phenyl-2-propenal)	[65]
	Geraniol ((2E)-3,7-dimethylocta-2,6-dien-1-ol)	[66]
	Limonene (1ne-methyl-4-(1-methyl phenyl))	[67]
	Farnesol (3,7,11-trimethyl-2,6,10-dodecatrien-1-ol)	[68]
	Florhydral (isopropylphenylbutanal)	[69]
	Linalyl acetate (bergamol, 3,7-dimethylocta-1,6-dien-3-yl acetate)	[70]
Deodorizing effect by natural products	*Eribotrya japonca*	[71]
	*Lotus corniculatus*	[72]
	*Salvia officinalis* (sage)	[73]
	*Rosmarinus officinalis* L. (rosemary)	[74]
	*Lavandular angustifolia* Mill. (lavender)	[75,76]
	Thyme herbs (*Thymus vulgaris* L. and *Thymus zygis* L.)	[77]

**Table 3 ijms-26-10415-t003:** Twenty-six allergenic fragrances according to the EU directive.

No.	Items	Chemical Structure	No.	Items	Chemical Structure
1	Amylcinnamal	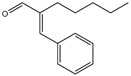	14	Farnesol	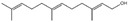
2	Amylcinnamyl alcohol	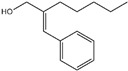	15	Geraniol	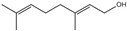
3	Anicyl alcohol(4-methoxybenzyl alcohol)	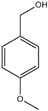	16	Hexyl cinnamicaldehyde(alpha-Hexylcinnamaldehyde)	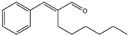
4	Benzyl alcohol	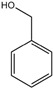	17	Hydroxyl-citronellal	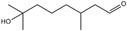
5	Benzyl benzoate	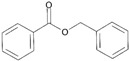	18	Hydroxy-methylpentyl-cyclohexenecarboxaldehyde	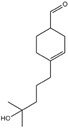
6	Benzyl cinnamate	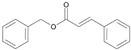	19	Isoeugenol	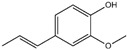
7	Benzyl salicylate	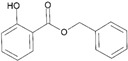	20	_D_-limonene	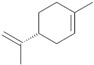
8	Cinnamyl alcohol	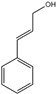	21	Linalool	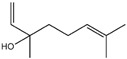
9	Cinnamal(Cinnamaldehyde)	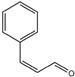	22	Methyl heptin carbonate	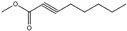
10	Citral	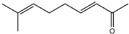	23	3-methyl-4-(2,6,6-tri-methyl-2-cyclohexen-1-yl)-3-buten-2-one(α-Cetone)	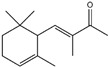
11	β-Citronellol	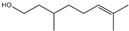	24	Oak moss and treemoss extract	-
12	Coumarin	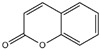	25	Treemoss extract	-
13	eugenol	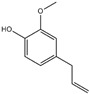	26	2-(4-tert-butylbenzyl) propionaldehyde(Lilial)	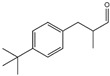

## Data Availability

The raw data supporting the conclusions of this article will be made available by the authors on request.

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
