# Peer review of "Human Body Malodor and Deodorants: The Present and the Future"

_ijms, 2025, doi:10.3390/ijms262110415_

Round 1

Reviewer 1 Report

Comments and Suggestions for Authors

My opinion is that the work is a very good overview of a very important topic that is not extensively or over studied. the text is of very good quality.

The novelty of the work is the focus on targeted intervention strategies—such as ABCC11 pump inhibition, bacterial active pump regulation, and AMRE blockade—which move beyond conventional deodorant approaches of masking odor or nonspecific antibacterial action. This precision-based perspective is of interest to readers as it opens new avenues for safer, microbiome-friendly, and more effective next-generation deodorant formulations. this review will increase the formulations and studies on new deodorant solutions

I would request just to rephrase the sentence 325 as it seems unclear.. has been observed...where?

Author Response

Reviewer 1’s Comments

My opinion is that the work is a very good overview of a very important topic that is not extensively or over studied. the text is of very good quality.

The novelty of the work is the focus on targeted intervention strategies—such as ABCC11 pump inhibition, bacterial active pump regulation, and AMRE blockade—which move beyond conventional deodorant approaches of masking odor or nonspecific antibacterial action. This precision-based perspective is of interest to readers as it opens new avenues for safer, microbiome-friendly, and more effective next-generation deodorant formulations. this review will increase the formulations and studies on new deodorant solutions

I would request just to rephrase the sentence 325 as it seems unclear.. has been observed...where?

Ans) Thank you so much for the generous comment and I rephrased the sentence to make it more clear.

Reviewer 2 Report

Comments and Suggestions for Authors

In this review manuscript, the authors presented an overview of human body malodor including its definition, causes, and mechanisms of formation and generation, and then mainly elucidated deodorants and antiperspirants including their ingredients and working principles, and finally analyzed and discussed the intervention strategy of next-generation deodorants according to current research and literatures. The topic in this study seems fine, however, the presentation looks more like a scientific introduction with a just simple collection of data/information together rather than a real review article to synthesize existing scholarly research on the selected topic, and to present a critical analysis of the current state of knowledge. Also, writing needs improvement with clear logic and clear expression to avoid confusion. Here are some concerns and suggestions for the authors’ consideration:         

  • Line 22, “However, next-generation deodorants have several adverse effects”. This is confusing as logically, next-generation deodorants should have no or less adverse effects. It’d better change to “current deodorants have several adverse effects”
  • The first paragraph in Section 1.2 Causes of Malodors, is very hard to understand the points, in particular, the statement “However, the significance of these compounds has declined, as the number of individuals for whom they are considered as a primary cause of axillary malodor has decreased to 50% or less”. Actually, this was misunderstood as the authors in the original citation (Ref. 11) meant that androstenone 1 is only perceived by 50% of human population but not like only 50% or less of individuals for whom they are considered as primary cause od axillary malodor. The points in this paragraph were not expressed clearly and logically.
  • Figure 1, the ‘sh’ in ‘PepTsh’ should be subscript to match with other same names in the text, like the one in line 116
  • Line 164, ‘… and an antiperspirant, which is a deodorant, functions by …’ states that an antiperspirant is a deodorant. However, it seems that antiperspirants are different from deodorant as they function in different ways. It’d better to delete it.
  • Many names of odorless precursors and malodor metabolites were mentioned in Section 1, however, there were no structures, which makes it hard to follow and understand the points stated in the text, especially, this section is to introduce the cause of malodors and their generation mechanisms
  • It would be good to include some illustrative diagrams to explain the working principles of deodorants and antiperspirants and demonstrate the intervention strategy of next-generation deodorants.
  • Most of the discussion section just repeated the information that was introduced in previous sections rather than give more prospective analysis in the field.   
Comments on the Quality of English Language

In this review manuscript, the authors presented an overview of human body malodor including its definition, causes, and mechanisms of formation and generation, and then mainly elucidated deodorants and antiperspirants including their ingredients and working principles, and finally analyzed and discussed the intervention strategy of next-generation deodorants according to current research and literatures. The topic in this study seems fine, however, the presentation looks more like a scientific introduction with a just simple collection of data/information together rather than a real review article to synthesize existing scholarly research on the selected topic, and to present a critical analysis of the current state of knowledge. Also, writing needs improvement with clear logic and clear expression to avoid confusion. Here are some concerns and suggestions for the authors’ consideration:         

  • Line 22, “However, next-generation deodorants have several adverse effects”. This is confusing as logically, next-generation deodorants should have no or less adverse effects. It’d better change to “current deodorants have several adverse effects”
  • The first paragraph in Section 1.2 Causes of Malodors, is very hard to understand the points, in particular, the statement “However, the significance of these compounds has declined, as the number of individuals for whom they are considered as a primary cause of axillary malodor has decreased to 50% or less”. Actually, this was misunderstood as the authors in the original citation (Ref. 11) meant that androstenone 1 is only perceived by 50% of human population but not like only 50% or less of individuals for whom they are considered as primary cause od axillary malodor. The points in this paragraph were not expressed clearly and logically.
  • Figure 1, the ‘sh’ in ‘PepTsh’ should be subscript to match with other same names in the text, like the one in line 116
  • Line 164, ‘… and an antiperspirant, which is a deodorant, functions by …’ states that an antiperspirant is a deodorant. However, it seems that antiperspirants are different from deodorant as they function in different ways. It’d better to delete it.
  • Many names of odorless precursors and malodor metabolites were mentioned in Section 1, however, there were no structures, which makes it hard to follow and understand the points stated in the text, especially, this section is to introduce the cause of malodors and their generation mechanisms
  • It would be good to include some illustrative diagrams to explain the working principles of deodorants and antiperspirants and demonstrate the intervention strategy of next-generation deodorants.
  • Most of the discussion section just repeated the information that was introduced in previous sections rather than give more prospective analysis in the field.   

Author Response

Reviewer 2’s Comments

In this review manuscript, the authors presented an overview of human body malodor including its definition, causes, and mechanisms of formation and generation, and then mainly elucidated deodorants and antiperspirants including their ingredients and working principles, and finally analyzed and discussed the intervention strategy of next-generation deodorants according to current research and literatures. The topic in this study seems fine, however, the presentation looks more like a scientific introduction with a just simple collection of data/information together rather than a real review article to synthesize existing scholarly research on the selected topic, and to present a critical analysis of the current state of knowledge. Also, writing needs improvement with clear logic and clear expression to avoid confusion. Here are some concerns and suggestions for the authors’ consideration:         

Line 22, “However, next-generation deodorants have several adverse effects”. This is confusing as logically, next-generation deodorants should have no or less adverse effects. It’d better change to “current deodorants have several adverse effects”

Ans) Thank you so much for the generous comments and I amended the points according to the comments.

The first paragraph in Section 1.2 Causes of Malodors, is very hard to understand the points, in particular, the statement “However, the significance of these compounds has declined, as the number of individuals for whom they are considered as a primary cause of axillary malodor has decreased to 50% or less”. Actually, this was misunderstood as the authors in the original citation (Ref. 11) meant that androstenone 1 is only perceived by 50% of human population but not like only 50% or less of individuals for whom they are considered as primary cause od axillary malodor. The points in this paragraph were not expressed clearly and logically.

Ans) I appreciated that one of authors in my cited references reviewed my manuscript and I amended the sentence according to the comments.

Figure 1, the ‘sh’ in ‘PepTsh’ should be subscript to match with other same names in the text, like the one in line 116.

Ans) Thank you so much and I amended the Figure 1.

Line 164, ‘… and an antiperspirant, which is a deodorant, functions by …’ states that an antiperspirant is a deodorant. However, it seems that antiperspirants are different from deodorant as they function in different ways. It’d better to delete it.

Ans) Thank you so much and I amended the sentence according to the comments. And I changed the subtitle from ‘2. Deodorants and Antiperspirants’ to 2. Deodorants.

Many names of odorless precursors and malodor metabolites were mentioned in Section 1, however, there were no structures, which makes it hard to follow and understand the points stated in the text, especially, this section is to introduce the cause of malodors and their generation mechanisms

Ans) Thank you so much for the informative comments and to make the readers’ clear understanding I added Table1. Odorless precursors and odoriferous metabolites.

It would be good to include some illustrative diagrams to explain the working principles of deodorants and antiperspirants and demonstrate the intervention strategy of next-generation deodorants.

Ans) Thank you so much for nice suggestion and I make the illustrative diagram to the next-generation deodorants. And I put the Figure into the manuscript as Figure 2. Diagram of the intervention strategy of next-generation deodorants.

Most of the discussion section just repeated the information that was introduced in previous sections rather than give more prospective analysis in the field. 

Ans) Thank you so much for the comment and I amended the Discussion section based on the comment.

Reviewer 3 Report

Comments and Suggestions for Authors

This manuscript mainly summarizes the mechanism of human axillary malodor formation, the currently used malodor and deodorants, and the intervention strategy of next-generation deodorants, providing certain insights for the application and development of human body malodor and deodorants. Some modifications should be considered before publication.

  1. Line 22, please verify whether it should be “next-generation” or “now”, and briefly outline the adverse effects.
  2. Line 18, please check “efflux of malodorous metabolites into the axilla via axillary malodor-releasing enzymes (AMREs)”. According to the manuscript, the role of AMREs is to catalyze the production of malodorous substances from odorless precursors, instead of transporting these substances into the axilla.
  3. Line 41, as Staphylococcus sp. and Corynebacterium sp. are considered normal flora in humans, please explain the differences in the distribution of these bacterial communities in individuals with or without malodor.
  4. Lines 54/58,please check the word“cleaved. Is it “converted” or “fermented”?
  5. Section 1.2, the logic of this section is weak. It is recommended to include a summary to outline the aspects of malodor formation, such as components, gene expression, diet, age, and gender, to help readers understand more quickly.
  6. Lines 133-136, please add content to enhance the logical connection.
  7. Section 2.2.1, could you please add the molecular mechanisms of antiperspirants?
  8. Line 192, “However, to date, the relation-ship between aluminum ingredient usage and breast cancer occurrence is unclear” is recommended to delete.
  9. Is it possible to include the molecular mechanisms of eliminating ammonia odor?
  10. Table 2, the chemical structures are not clearly presented. It is suggested to use professional software such as ChemDraw to redraw them.
  11. Line 222, could you supplement the molecular mechanisms of “eliminate ammonia odor”?
  12. Section 2.2.3, a table as that in Section 2.2.2 was recommended for easier understanding.
  13. Line 325,does “significant difference” refer to differences between individuals?
  14. Sections 3.1/3.2, it is suggested to first introduce the mechanisms and then provide examples to create better coherence.

Author Response

Reviewer 3’s Comments

This manuscript mainly summarizes the mechanism of human axillary malodor formation, the currently used malodor and deodorants, and the intervention strategy of next-generation deodorants, providing certain insights for the application and development of human body malodor and deodorants. Some modifications should be considered before publication.

  1. Line 22, please verify whether it should be “next-generation” or “now”, and briefly outline the adverse effects.

Ans) Thank you so much for the generous comment and I changed the word to current.

  1. Line 18, please check “efflux of malodorous metabolites into the axilla via axillary malodor-releasing enzymes (AMREs)”. According to the manuscript, the role of AMREs is to catalyze the production of malodorous substances from odorless precursors, instead of transporting these substances into the axilla.

Ans) Thank you so much and to clear the function of AMREs I amended the word from via to after converted by.

  1. Line 41, as Staphylococcus and Corynebacteriumsp. are considered normal flora in humans, please explain the differences in the distribution of these bacterial communities in individuals with or without malodor.

Ans) They are normal flora but in the malodor generation sites such as axilla and foot if the appropriate condition can be set to proliferate them such as humidity, nutrients, and temperature they can convert malodor from odorless precursor using with AMREs.

  1. Lines 54/58,please check the word“cleaved. Is it “converted” or “fermented”?

Ans) Thank you so much for the question but 3 words were used as same meaning because bacterial fermentation implies to cleave somethings. However, according to the comment and to avoid the readers’ misunderstanding I changed “cleaved” to “fermented”.

  1. Section 1.2, the logic of this section is weak. It is recommended to include a summary to outline the aspects of malodor formation, such as components, gene expression, diet, age, and gender, to help readers understand more quickly.

Ans) Thank you so much for the informative comment and I added the subtitles according to the comment.

  1. Lines 133-136, please add content to enhance the logical connection.

Ans) Thank you so much for the comment and I amended the sentences based on the causal chain.

  1. Section 2.2.1, could you please add the molecular mechanisms of antiperspirants?

Ans) The molecular mechanisms of antiperspirants is same as the purpose of antiperspirants to inhibit secretion of sweat from sweat gland pores via gel plugging. And I mentioned that before explaining each antiperspirant.

  1. Line 192, “However, to date, the relation-ship between aluminum ingredient usage and breast cancer occurrence is unclear” is recommended to delete.

          Ans) Thank you so much for the comment and I deleted the sentence.

  1. Is it possible to include the molecular mechanisms of eliminating ammonia odor?

Ans) Thank you so much the informative question and I add the molecular mechanism of that.

  1. Table 2, the chemical structures are not clearly presented. It is suggested to use professional software such as ChemDraw to redraw them.

Ans) Thank you so much and I changed the chemical structures to clearly understand them.

  1. Line 222, could you supplement the molecular mechanisms of “eliminate ammonia odor”?

Ans) Thank you so much and I add the molecular mechanism of that.

  1. Section 2.2.3, a table as that in Section 2.2.2 was recommended for easier understanding.

Ans) Thank you so much and I mentioned in Section 2.2.2.

  1. Line 325,does “significant difference” refer to differences between individuals?

Ans) Yes. To clearly explain the meaning to that I added the word.

  1. Sections 3.1/3.2, it is suggested to first introduce the mechanisms and then provide examples to create better coherence.

Ans) Thank you so much. According to the above comment I arranged the pharagraphs.
